# *Hypebaeus cooteri* sp. nov., the Nemoral Species of Soft-Winged Flower Beetles (Coleoptera, Malachiidae) in North Asia

**Sergei E. Tshernyshev** [1,2]

1 Institute of Systematics and Ecology of Animals, Siberian Branch of the Russian Academy of Sciences, Funze Str. 11, 630091 Novosibirsk, Russia; sch-sch@mail.ru
2 Tomsk State University, Lenina Prospekt, 36, 634050 Tomsk, Russia

**Abstract:** A new species of soft-winged flower beetle, *Hypebaeus cooteri* Tshernyshev, sp. nov., external appearance very close to *H. flavipes* (Fabricius, 1787) but differs in shape of the elytral appendages and dark colouration of apical margin of the elytra is described from the Russian Far East. Phenomenon of disjunctive distribution of the new species from the main areal of the genus is discussed on the basis of nemoral faunogenesis during the Late Pleistocene and the Middle Holocene in dependence of broad-leaf forest distribution. Oak trees are considered as typical habitat of the genus *Hypebaeus* Kiesenwetter, 1863 on the basis of field study by Jonathan Cooter in Moccas Park National Nature Reserve on old oak trees in UK. *H. flavipes*, the nominative species of the genus *Hypebaeus*, is redescribed from the specimens collected in Great Britain. Illustrations of external appearance and special male characters for both species are provided.

**Keywords:** *Hypebaeus*; areal disjunction; new species; North Asia; Southern Primorie; taxonomy





## 1. Introduction

The genus *Hypebaeus* Kiesenwetter, 1863 belongs to the tribe Ebaeini of the subfamily Malachiinae and includes soft-winged flower beetles of small sizes (2–3 mm in length) with thin and short extremities, simple filiform antennae, and in males also with simple legs lacking comb above the 2nd tarsomere in anterior tarsi and elytra impressed apically and provided with ear-shaped appendages compactly closing apices of elytra.

Amongst the congeners of the tribe, it is closer to nominative genus *Ebaeus* Erichson, 1840, in which males are also provided with compactly closing appendages in elytral apices, but all of them are provided with a comb above the second segment of anterior tarsi.

Similarity of appearance in a number of species has led to incorrect interpretation of the Palaearctic species of the tribe, when generic attribution has been determined on the basis of presence or absence of the only character, the tarsal comb in males. During this period of time, in the genus *Hypebaeus* were included newly describing species in which males possessed not compactly closing appendages in elytral apices but spicular processes or elytral folds. For improvement of taxonomic structure of the tribe, Walter [1] prepared the first generic revision. He divided all genera into two groups by the character of bilaciniate or undivided apical tergite (pygidium) in male. Both *Ebaeus* and *Hypebaeus* were placed into a group with undivided pygidium. The species with other shape of elytral apices and appendages, as was mentioned above, were attributed to new genera. Amongst species occurring in North Asia, species with spicular appendages in elytral apices of male belong to the genus *Hypomixis* Wittmer, 1995, and those species which elytral apices lack appendages but provided with complicatedly folded apical edges, belonging to the genus *Kuatunia* Evers, 1949 both genera lack tarsal comb in male.

A problem of re-examination of all species previously described without detail analysis of elytral apices structure in male, mainly in the genus *Hypebaeus* appeared necessary after new taxonomic division of the tribe Ebaeini, and remains actual at present time.

Currently, only nine species of true *Hypebaeus* are known from European part of the continent, of which only one species, *H. flavipes* was mentioned for North Asia [2–5]. The new species from Southern Primorie with no doubt belongs to *Hypebaeus*, and very close to *H. flavipes* by external appearance, but differs in shape of the elytral appendages and dark colouration of apical margin of the elytra. Detailed study of these specimens and comparison with *H. flavipes* from England, kindly presented by Jonathan Cooter, clearly showed that this species is not *H. flavipes*, but independent species which is described below as new, *Hypebaeus (Hypebaeus) cooteri* Tshernyshev, sp. nov.

Phenomenon of disjunctive distribution of the new species from the main areal of the genus is discussed on the basis of nemoral faunogenesis during the Late Pleistocene and the Middle Holocene in dependence of broad-leaf forest distribution is discussed. Oak trees are considered as typical habitat of the genus *Hypebaeus* on the basis of field study by Jonathan Cooter in Moccas Park National Nature Reserve on old oak trees in UK. Elsewhere in Europe *H. flavipes* has been found associated with Beech and Hornbeam [6].

## 2. Material and Methods

For descriptions, special male structures and genitalia were studied; "special male structures" term refers here to the impressed elytral apices and ear-shaped outer appendages, compactly close the elytral impressions.

This term is not analogue to the term "Excitatoren", which are different kinds of structures located in different parts of the male body of soft winged flower beetles and bearing ducts of pheromone glands necessary for female attraction and successful copulation [7–10]. The "special male structures" includes all typical parts of male irrespective of pheromone glands or not.

Illustrations have been prepared using specimens from the type locality: *Hypebaeus (Hypebaeus) cooteri* Tshernyshev, sp. nov., holotype, male—Russia, Khabarovskii Krai, Botchinsky Nature Reserve; *Hypebaeus (Hypebaeus) flavipes* (Fabricius, 1787), male—Great Britain, England, Herefordshire, Moccas Park National Nature Reserve.

The specimens are currently deposited in the following institution, which is subsequently referred to by the acronym:

EATB = Federal Scientific Center of the East Asia Terrestrial Biodiversity, Far Eastern Branch of Russian Academy of Sciences, Vladivostok, Russia.

OUMNH = Oxford University Museum of Natural History, Great Britain.

SCH_ISEA = author's collection, which is kept in the Institute of Animal Systematics and Ecology, Siberian Branch of the Russian Academy of Sciences, Novosibirsk, Russia.

ZISP = collection of the Zoological Institute of the Russian Academy of Sciences, Saint Petersburg, Russia.

The beetles were studied using an Amscope trinocular stereomicroscope (Ultimate Trinocular Zoom Microscope 6.7X-90X Model ZM-2TY), and digital photographs were taken using a Carl Zeiss Stemi 2000 trinocular microscope and the AxioVision programme. Male genitalia, embedded in DMHF (dimethyl hydantoin formaldehyde), were mounted onto a transparent card and pinned under the specimen.

This article is registeresd in ZooBank under the link http://zoobank.org/urn:lsid: zoobank.org:pub:C99A95AB-8AAE-4F04-ACEA-1E1F786422D9.

## 3. Taxonomy

Class Insecta Linnaeus, 1758
Order Coleoptera Linnaeus, 1758
Suborder Polyphaga Emery, 1886
Superfamily Cleroidea Latreille, 1802
Family Malachiidae Fleming, 1821
Subfamily Malachiinae Fleming, 1821
Tribe Ebaeini Portevin, 1931

*Hypebaeus (Hypebaeus) cooteri* Tshernyshev, **sp. nov.** Figure 1H–O, 2 (square). http://zoobank.org/urn:lsid:zoobank.org:act:1569035D-6E57-4001-9D6C-BCA68B87ED02.

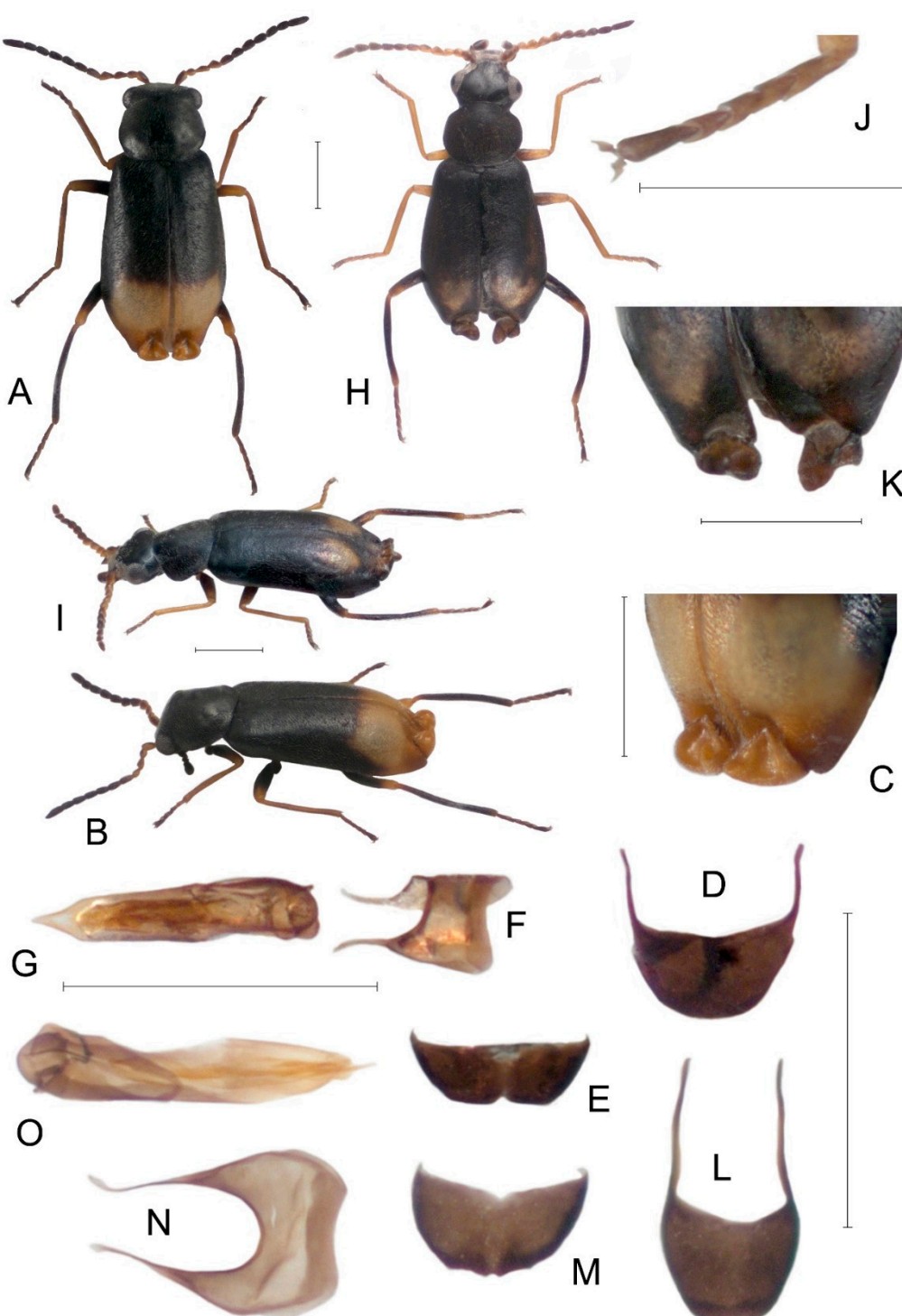

**Figure 1.** *Hypebaeus (Hypebaeus) flavipes* (Fabricius, 1787), (male) (**A–G**) and *Hypebaeus (Hypebaeus) cooteri* Tshernyshev, sp. nov. (holotype, male) (**H–O**). (**A,B,H,I**) External appearance. (**C,K**) Elytral apices. (**D,L**) Pygidium (apical tergite). (**E,M**) 8th ultimate abdominal ventrite (apical sternite). (**F,N**) Tegmen. (**N,O**) Aedeagus. (**J**) Left anterior tarsus. Scale bars 0.5 mm.

*Material.* Holotype, male: **Russia,** *Khabarovskii Krai:* Botchinsky Nature Reserve, right bank of Mulpa river, near Komarov kluch river, h~220 m above see level, 48°16′35″ N;

139°28′74″ E, 21.vi.2016, R. et E. Dudko leg. (SCH_ISEA); paratypes: ***Primorskii Krai:*** Sikhote-Alinsky Nature Reserve, Dalnegorskii raion, upper stream of Dzhigitovka river, Kabanii brook, 25.vi.2019, M. Sergeev leg.—6♂♂(OUMNH, SCH_ISEA, EATB); idem, cordon Maisa, 14–15.v.2001, A. Kirejtchuk leg.—1♂(ZISP); Anuchinskii Raion, Muraveika vill., 43°50′ N; 133°13′ E, 9.vi.1978, G. Lafer leg.—1♂(EATB); east foothill of Oblachnaya mountain, 43°41′ N; 134°11′ E, 10.vii.1977, G. Lafer leg.—2♀♀(EATB); Chuguevskii Raion, military base, from the wall of "banya" (bath house), 44°5′ N; 133°52′ E, 29.vi.1978, N. Moroz leg.—1♀(EATB); Lazoskii State Rezerve, America cordon, h~350 m a.s.l., 43°16′ N, 134°03′ E, 23–31.v.2009, S.A. Kurbatov leg.—3♂♂(ZISP); Lysaya Mountain, 45°47′ N, 136°30′ E, 30.vi.2020, M. Sergeev leg.—1♀(SCH_ISEA).

*Diagnosis.* The new species belongs to the subgenus *Hypebaeus (Hypebaeus)* and is close to *H. flavipes* which differs in shape of the elytral appendages and dark colouration of the elytra.

*Etymology.* The species is named in honour of outstanding English entomologist and coleopterologist, Jonathan Cooter, my dear friend, who kindly presented valuable data on *Hypebaeus flavipes* morphology and bionomy which allowed us to understand the independent status of new species from Southern Primorie.

*Description.* Male (Figure 1H,I). Head, pronotum, scutellum and elytra excepting triangular yellow spots on apical quarter, body of external appendages which edged with black-brown striae, outer sides of antennomeres from 6 to 10, palpi, posterior femora and tibiae excepting yellow comissure part, basal half of anterior and intermediate femora and almost completely all underside black-brown lacking metallic luster, remaining parts yellow; vesicles yellow-brown, thoracic mesepimere black-brown.

Body small, oval, evenly expanded just behind the humeri, maximum width at about apical quarter, oval, evenly rounded posteriorly and truncate at apices which are impressed and bear two ear-shaped appendages.

Head small, narrow, not elongate, not narrower than pronotum, interocular area flat, slightly impressed, dull, eyes small, round; surface densely finely punctate lacking microsculpture, evenly covered with thin short goldish adpressed hairs. Antennae attached near lateral side of the clypeus, clypeus transverse and narrow, labrum twice as wide as clypeus, transverse; palpi short, apical palpomere rectangular with rounded angles, slightly widened and flattened apically and obliquely cut at apex, look enlarged, intermediate and basal palpomeres small, transverse. Antennae filiform with slightly triangular intermediate antennomeres, long, almost reaching middle of the elytra; the 1st segment oval and enlarged, twice as long as the 2nd, the 2nd antennomere oval, the 3rd slightly elongate and subtriangular, the 4th antennomere wide triangular, 1.5 times as wide as the 2nd, 5th–10th antennomeres elongate-triangular, the same length as the 4th, apical antennomere is 1.5 times as long as the 10th, elongate, oval and pointed; antennomeres covered with pale-yellow fine, short erect and short adpressed hairs.

Pronotum transverse, evenly rounded at anterior and posterior anglers, not narrowed at base, with prominent anterior side and almost straight posterior one, disk weakly convex lacking depressions; margination very thin and more distinct on basal side; surface dull, sparsely finely punctured lacking microsculpture, evenly covered with short fine goldish adpressed hairs.

Scutellum distinct, narrow, longitudinal, rectangular at apex, flat, not marginate; the surface dull, distinctly punctured, sparsely covered with short adpressed goldish hairs.

Elytra oval, expanded at about apical third, apices obliquely truncate roundly impressed and with ear-shaped outer appendages, the outer edge of elytral apices slightly stretched into rounded horn. Appendages (Figure 1K) transverse, with dilated outer edge curved inside in outer and inner lateral sides, look rather rounded than angular; humeri small, weakly protruding, epipleurae narrow, distinctly marginate, suture thin with indistinct margination; surface weakly shining, densely punctured, punctures smooth, microsculpture indistinct, evenly covered with short goldish recumbent pubescence.

Wings well developed.

Legs simple, long and thin, posterior femora extending beyond elytral apices, posterior tibiae slightly curved outwards and widened posteriorly, covered with short adpressed goldish hairs; femora somewhat wider than tibiae, compressed; all tarsi 5-segmented, simple, lacking combs or appendages (Figure 1J), narrow and slightly elongate, 1st–4th tarsomeres slightly compressed, the apical tarsomere depressed, 4th tarsomere is shortest in all legs and is twice shorter than the 5th tarsomere, 1st and 5th tarsomere almost completely equal length in all legs, 2nd–3rd tarsomeres nearly equal in length and size, and their general length equal the length of the 1st tarsomere; claws short, thin and sharp, with dilated and rounded base.

Ventral body surface weakly shining, sparsely and indistinctly punctured, sparsely covered with fine, adpressed goldish pubescence; metathorax convex, simple, lacking appendage or hair tuft; pygidium (apical tergite) (Figure 1L) simple, evenly narrowed apically and looks equilateral; 8th ultimate abdominal ventrite (apical sternite) (Figure 1M) wide and transverse, 1.9 times as wide as long, evenly narrowed and rounded apically and with complicatedly truncate distal side: small round emargination is in a middle, and two semi-round emarginations are located on left and right sides of the central one; tegmen short and wide, transverse, 1.7 times longer than wide, widely emarginate in the middle and with short subrounded base and thin long and slightly curved appendages (Figure 1N); Phallus (Figure 1O) simple, moderately curved dorsally, pointed anteriorly, with a sharp small and not elongated apical lamella, penis roundly widened in apical third, inner sack with two long slightly curved denticles near apex, and a pair of denticles ant base.

Length (holotype) 2.0 mm, width (at elytral base) 0.5 mm.

Female differs by simple elytra lacking impressions and appendages, oval and strongly widened posteriorly body, completely black colouration of elytra lacking yellow spots, antennae more slender and shorter, extending beyond the base of elytra, only 1st–4th antennomeres yellow, remaining antennomeres brown, 1st antennomere not enlarged, oval, second antennomere round, 4th triangular and distinctly elongate, tarsi thin, simple, lacking protruding plates. Length 2.3 mm, width (at elytral base) 0.7 mm.

*Distribution.* Known only from Sikhote-Alin mountain range, Southern Primorie, the Russian Far East (Figure 2 (quadrates)).

*Hypebaeus (Hypebaeus) flavipes* (Fabricius, 1787). Figure 1A–G, 2 (circle).

*Material.* **Great Britain, England,** *Herefordshire:* Moccas Park National Nature Reserve 52°4′ N; 2°55′ W, beaten—ancient oak, 22.vi.1978, J. Cooter leg.—2♂♂, 2♀♀(SCH_ISEA).

*Redescription.* Male (Figure 1A,B). Head, pronotum, scutellum, and elytra excepting yellow apical third, antennomeres from 5 to 11, palpi, posterior femora and tibiae excepting yellow apical 1/5 of femora and comissure part of tibiae, basal half of anterior and intermediate femora and almost completely all underside black-brown lacking metallic luster, remaining part including mouth, palpi, mandibles, clypeus and labrum yellow; vesicles pale-yellow, thorax mesepimere black-brown.

Body small, weakly oval almost parallel-sided, slightly expanded at about apical quarter, evenly rounded posteriorly and cut straight at apices which are impressed and bear two ear-shaped appendages.

Head small, narrow, not elongate, not narrower than pronotum, interocular area flat, very weakly impressed, slightly shiny, eyes small, round; surface densely finely punctured with distinct microsculpture, evenly covered with thin short dark adpressed hairs. Antennae attached near anterior lateral edge of the clypeus, clypeus transverse and narrow, labrum 1.5 times as wide as clypeus, transverse; palpi short, apical palpomere rectangular with rounded angles, slightly widened and flattened apically and slightly obliquely cut at apex, not noticeably enlarged, intermediate and basal palpomeres small, transverse. Antennae filiform with oval and slightly triangular antennomeres, long, almost reaching middle of the elytra; the 1st antennomere conic, widened apically, the 2nd antennomere round, twice as short as the previous and narrower, the 3rd and the 4th antennomeres elongate and triangular, slightly elongate and subtriangular, the 4th antennomere wide triangular, 1.3 times as wide as the 2nd, 5th–10th antennomeres oval, the same length as

the 4th but narrower, apical antennomere is twice as long as the 10th, elongate, oval and evenly narrowed and rounded apically; antennomeres sparsely covered with pale, fine, short erect hairs.

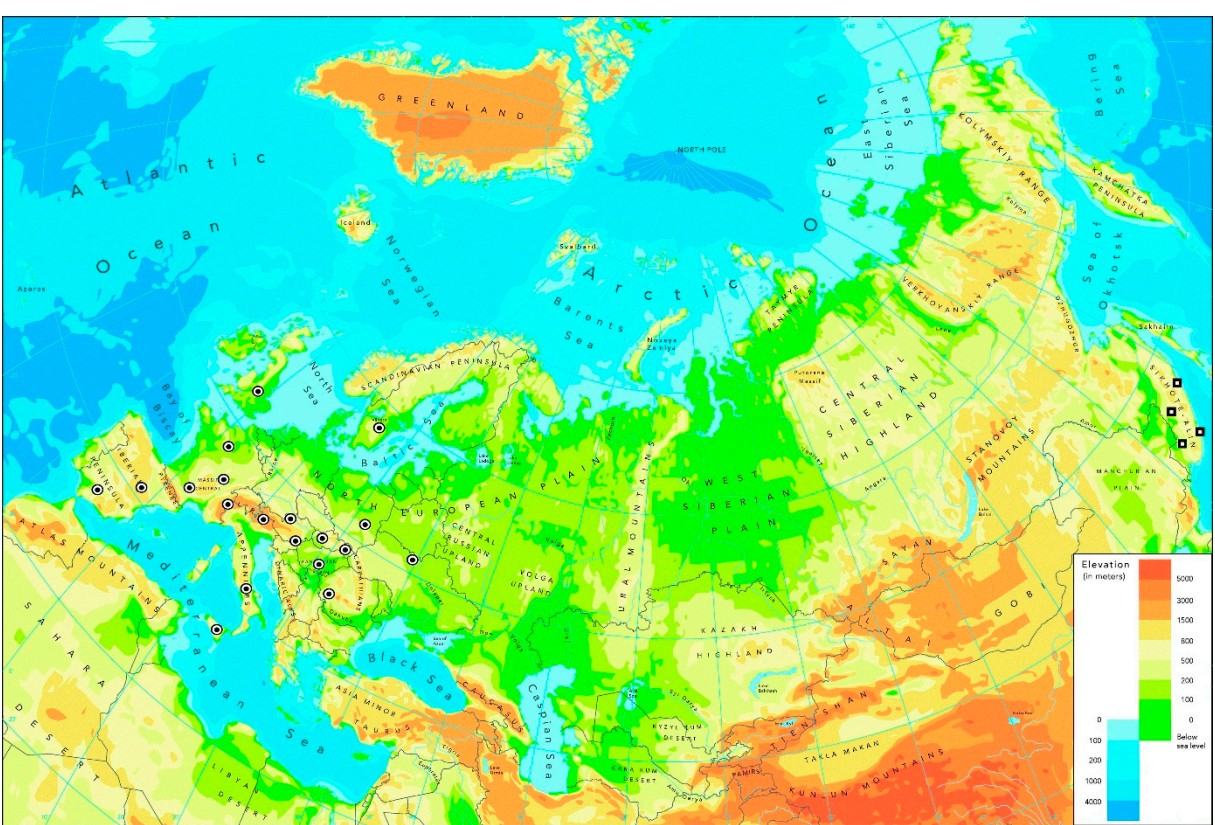

**Figure 2.** Distribution map of *Hypebaeus (Hypebaeus) flavipes* (Fabricius, 1787) (circles) and *Hypebaeus (Hypebaeus) cooteri* Tshernyshev, sp. nov. (quadrates): Source of the map: https://www.eea.europa.eu/legal/copyright. Copyright holder: European Environment Agency (EEA).

Pronotum transverse, evenly rounded at about anterior and posterior anglers, not narrowed at base, with prominent anterior border and almost straight posterior border, disk weakly convex lacking depressions; margination very thin and more distinct on basal and anterior sides; surface weakly shining, finely punctured with distinct microsculpture, covered with short fine dark adpressed hairs.

Scutellum distinct, narrow, weakly longitudinal, oval-rectangular at apex, flat, not marginate; the surface dull, weakly punctured, and sparsely covered with short adpressed dark hairs.

Elytra oval, weakly expanded at about apical quarter, apices truncate straight with round impressions and triangular ear-shaped outer appendages, the outer edge of elytral apices evenly rounded. Appendages (Figure 1C) transverse, regular triangle shape, with thin outer edge not curved inwardly; humeri small, not protruding, epipleurae narrow, slightly widening apically and weakly marginate, suture thin, lacking margination (Figure 1G); surface dull, densely and finely punctured, microsculpture distinct, evenly covered with short dark recumbent pubescence.

Wings well developed.

Legs simple, long and thin, posterior femora not reaching the elytral apices, posterior tibiae slightly curved outwards and widened posteriorly, covered with short adpressed hairs; femora somewhat wider than tibiae, compressed, tibiae thin; all tarsi 5-segmented, simple, lacking combs or appendages, narrow and slightly elongate, 1st–4th tarsomeres slightly compressed, the apical tarsomere depressed, 4th tarsomere is slightly shorter in all legs, the 5th tarsomere is the largest and is equal to general length of the 1st and the

2nd tarsomeres, 2nd–3rd tarsomeres nearly equal in length and size, and their general length equal to the length of the 1st tarsomere in posterior legs and longer in anterior and intermediate legs; claws short, thin and sharp, with round base.

Ventral body surface weakly shining, sparsely and indistinctly punctured, sparsely covered with fine, adpressed dark pubescence; metathorax convex, simple, lacking appendage or hair tuft; pygidium (apical tergite) (Figure 1D) simple, evenly narrowed apically, narrow, transverse; 8th ultimate abdominal ventrite (apical sternite) (Figure 1E) wide and transverse, 3.2 times as wide as long, evenly narrowed and cut apically, with narrow emargination in a middle, tegmen short and wide, transverse, 1.9 times longer than wide, slightly emarginate in the middle and with short subround base and thin short and pointed appendages (Figure 1F); Phallus (Figure 1G) simple, weakly curved dorsally, pointed anteriorly, with elongate narrow and apical lamella, penis not widened apically, inner sack with two long slightly curved along it hollow spines in a middle, and a group of small denticles are visible at apex.

Length 2.1 mm, width (at elytral base) 0.7 mm.

Female differs by simple elytra lacking impressions and appendages, round-oval and strongly widened posteriorly body, almost completely black colouration of elytra except small yellow spot at sutural angle of each elytron, antennae more slender and short, reaching the base of pronotum only, 1st antennomere not enlarged, oval, 3rd and 4th weakly triangular and distinctly elongate, tarsi more thin, all tarsomeres excepting the 5th provided with protruding pedal plates. Length 2.4 mm, width (at elytral base) 0.8 mm.

*Distribution.* Widely spread disjointed distribution in Central and Southern Europe, and known from Austria; Belgium; Bosnia and Herzegovina; Corsica, Croatia, Czech Republic; France; Germany; Hungary; Italy; Luxembourg; Poland; Russia; Spain; Sweden; Switzerland (Figure 2 (circles)).

## 4. Typical Habitat for *Hypebaeus*

In Eurasia Malachiidae beetles occur in various biotopes, preferring arid types of steppes and semi-deserts in intracontinental regions, while in Europe, located mainly in territory with wet and mild climatic conditions of Atlantic and Arctic oceans basins beetles prefer forest associations. Bionomy of European species is still poorly studied, and can be considered by analogy with data known from other regions, for example, with the Central Asian species of *Cephaloncus* Westwood, 1863 that was recorded under bark hunting small invertebrates including Buprestidae beetle larvae [11] or species inhabiting coastal biotopes in Japan described in a series of papers on Japanese malachids which larvae occurred on "mossy rocks along the stream, and larvae and prepupa in clefts of rocks" [12–19]. Thus, larvae of Malachiidae beetles are predators hunting small invertebrates under bark or on stems of trees.

Habitat of the nominative species of the genus *Hypebaeus*, the *H. flavipes* is currently studied by Jonathan Cooter in Moccas Park National Nature Reserve, UK, with ancient oak trees. Jon kindly provided beetle specimens, letters and photos of his study to compare to what was obtained from South Primorie. Detailed description and illustration of his specimens are given above, but a brief note on his study of the old oak trees as beetle habitat is: "The canopy investigation was arranged simply because *Hypebaeus* has only been looked from by entomologists standing on the ground reaching maybe 2 m. The very first UK specimens of *Hypebaeus* were actually found on a rainy day in 1934 by sweeping the grass under an ancient oak. It has never been swept since, always beaten from oak foliage. I've found it over a 45 year period and my experience indicated it (both sexes) prefer thin twigs growing near to or around cavities or rotholes in the most ancient oak trees" (Jonathan Cooter, personal communication). Old oaks provide necessary conditions for *Hypebaeus* larvae development, and, probably, the same is in the Far East of Russia. Not having studied life-cycle of *H. cooteri* sp. nov. it is obvious that all specimens were collected from or near oak trees, so this species should be considered as obligatory associated with oaks. Jon gave additional remark: "I found it [*H. flavipes*] on 17 of the most ancient oaks in

Moccas Park; these are estimated to be at least 400-years old, doubtless much older. There are about 300 ancient oaks in the Park and I have searched every one of these for *Hypebaeus*, but only found it on 17" (Jonathan Cooter, personal communication).

## 5. Discussion and History of the Species Origin

In fact, all species of the genus *Hypebaeus* are restricted in their distribution by European part of the continent, only a new species, *H. cooteri* sp. nov. occurs in easternmost locality of Asia. This phenomenon can be explained by characteristic bionomy of the *Hypebaeus* species associated with old oak trees in their life-cycle. Broad-leaf forests are typical for wet and mild climate of Europe, and are missing in the centre of the continent. Currently, only secondary plantations of single trees occur in South Siberia, several trees of Mongolian oak are grown in East Siberia, but are considered as crooked forest not reaching 100 years due to unsuitable climatic conditions in the region. In contrast, Southern Primorie is a comfortable place for oak growth with the moist and mild climate more similar to that typical of Europe. Old trees with wide crown and thick trunk are typical for the regional forests and grow there for many years. They present the ideal habitat for a number of Malachiidae species. The phenomenon of disjunction range of the genus *Hypebaeus* can be explained on the basis of the thesis of nemoral faunogenesis presented by Vladimir Dubatolov and Oleg Kosterin [20]. The authors has analysed distribution of some lepidopteran species associated with broad-leaf forests and compare their occurrence with historical spread of these forests on the basis of pollen presence in fossil deposits in Eurasia. It was shown that the full belt of broad-leaf forest occupied the Palaearctic in Pleistocene, and later collapsed and vanished in the centre of the continent. In the Quaternary period, the forests renewed there several times during the Late Pleistocene and the Middle Holocene. This was good opportunity to study the species spread over the broad-leaf belt throughout Palearctic from West to East. Thus, *H. flavipes* could spread over the continuous belt of broad-leaf forests with oaks to the Far East. Then, when the belt disappeared in Siberia during the Subboreal period of the Holocene, the species was isolated in Southern Primorie, forming an independent species different from all European congeners of *Hypebaeus*. This is one of the possible scheme of origin of *H. cooteri* sp. nov.

Thus, the new species, *H. cooteri* sp. nov., found in Southern Primorie in the easternmost locality of the continent with disjunction of the genus distribution in central part, presents new evidence for conception of nemoral faunogenesis in Eurasia proposed by Dubatolov & Kosterin [20] on the basis of Lepidoptera.

The discussed phenomenon of remote disjunctive range of the genus most species of which are distributed strictly in Europe, does not change conception on the main areal of the genus. In fact, autochthon centre of the genus with a number of species distributed in different regions is located in Europe. A new species, which is very close by external appearance to typical representative of the genus the nominative *H. flavipes*, just present evidence of wide distribution of the genus in the past. There is no evidence that other species of *Hypebaeus* can be found in North or East Asia due to strongly restricted area of oak tree distribution established during the Subboreal period of the Holocene changes of flora.

Broad-leaf forests of the Far East are inhabited by a number of unique relic and endemic species. In Malachidae it is not only *H. cooteri* sp. nov., but *Malachius glaucoviolaceus* Tshernyshev, 2009, *Cordylepherus pseudofaustus* Tshernyshev, 2009 and *Ebaeus legalovi* Tshernyshev, 2009 [5,21]. All of them are described from oak forests or nearby and are known from the Russian Far East only. Probably, broad-leaf forests were refuges for nemoral fauna inhabited Far East in the past during climatic changes of Pleistocene and Holocene.

**Funding:** This study was held under the Program of Basic Scientific Research (FNI) of the State Academies of Sciences, project No. 122011800267-4.

**Institutional Review Board Statement:** Not applicable.

**Data Availability Statement:** Not applicable.

**Acknowledgments:** The author is especially grateful to Jonathan Cooter (Oxford University Museum of Natural History, UK) for the kind help with the *Hypaebeus* bionomy in UK, *H. flavipes* specimens presented for study, valuable advice and final correction of the manuscript. The author is grateful to all colleagues who collected and presented the specimens of *H. cooteri* sp.n. for the study, namely: Alexander Kirejtchuk (Zoological Institute. RAS, St.-Petersburg), Sergei Kurbatov (All-Russian Plant Quarantine Center (VNIIKR), Moscow), Roman ans Evgenii Dudko (ISEA, Novosibirsk), Maksim Sergeev (East Asia Terrestrial Biodiversity Scientific Center, Vladivostok). The author cherishes the memory of and the deepest gratuity of late Viktor Nikolaevich Kuznetsov, who, with the kindness to people that was so typical of him, handed Malachiidae collection of BPI FEB RAS (now East Asia Terrestrial Biodiversity Scientific Center, Vladivostok) during his last visit to Novosibirsk in 2008 to study.

**Conflicts of Interest:** The authors declare no conflict of interest.

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
