# Peer review of "Hypebaeus cooteri sp. nov., the Nemoral Species of Soft-Winged Flower Beetles (Coleoptera, Malachiidae) in North Asia"

_diversity, doi:10.3390/d14100875_

Round 1

Reviewer 1 Report

Dear Editor, 

This manuscript presents a new species of beetle, which is compared with a morphologically close European species. 

In general, the English of the manuscript should be carefully revised. This is because the author uses complex sentences and grammatical errors that make it difficult to read. I am not a native English speaker, but I consider that it should be proofread by a native speaker. I have made many corrections and suggestions in the text.

A conclusion to this manuscript and perspectives are needed. The figures should be rearranged and the numbering should be well designated. 

I can't judge by the characters used for the description of the species, because I am not an expert on this family. 

The manuscript may be considered for publication in this journal after further revision.

Author Response

Dear colleague,

thank you very much for precise review of my manuscript and kind recommendation to improve the text. English grammar of the revised version of manuscript was read and corrected by Jon Cooter, all necessary changes have been implemented. Thank you again! Sergei 

Reviewer 2 Report

Dear author,

The English language is not appropriate for publication. Please send the manuscript to a native English speaker. I support that J. Cooler will help you as its seems that He Supports your studies.

Author Response

(The authors gave the same response as above.)

Round 2

Reviewer 1 Report

Dear editor,

This reviewed version of the manuscript submitted by the author has only a few alterations and almost nothing of my comment and recommendations had taken into the account.  So for me, it is not worth re-evaluating this version of the manuscript. The answer of the author was only to review the English but there are no answers to my questions and comments. The most important thing to consider is that there is no conclusion and the legend of Fig. 1 does not match with the images and the text. I suggest that the author having into account some of the suggestions I did in the anterior version of the manuscript. 

Best regards, 

Author Response

Dear colleague,

thank you very much for your time and valuable comments on the manuscript. Sorry, I have missed several stupid mistakes, such as wrong figure numerartion under the legends to drawings. Most of highlighted places are corrected, but style of the text in the Introduction is remaind in author's  version, which was revised and proved by Jon Cooter as native English-speaker. 

Thanks again!

With the kindest wishes,

Sergei

Reviewer 2 Report

There are still mistakes in the English language, etc.   Line 31: has been determined   Otherwise it is ok.

Author Response

Dear colleague,

thank you very much for your time and valuable comments on the manuscript.

With the kindest wishes,

Sergei

Round 3

Reviewer 1 Report

Dear author, 

I recommend you organize the results as I suggested to you previously. In a taxonomical description, the first things that must appear after the designation of the species name are - Type material, - Diagnosis, - Description (starting with the body size), - Etymology, - Distribution ...

*it is not necessary to call the authorship of the species/genus name each time. It is necessary for the text only the first time that this is mentioned. 

Line 60 Hypebaeus -> in italic 

Line 197: to H. flavipes you didn't have a description you re-described the species 

Line 304: This phenomenon is explainable ... *Change by: "This phenomenon can be explainable ..."

Line 326:  This is one of the most realistic scheme of ... *Change by:  "This is one of the possible scheme of..."

Line 339: no evidence to suppose any other species ... *Change by "no evidence that other species ..."

Line 350 and 353: You have to use the author (in 3rd person) or I (1st person), but not both.

Item 5. I considered this item need be to move to the discussion. As you mention in the abstract and introduction ".. the genus is discussed on the basis of nemoral faunogenesis during the Late Pleistocene and the Middle Holocene "  . And because it is really a discussion based in your results. 

Author Response

Dear colleague,

thank you again for further revision of my manuscript. All your recommendations are implemented in the text excepting position of beetle size data. In all my publications this character is given in conclusion of the description and this new version could confuse readers accustomed the previous order.

Thank you again
